# Gaussian Membership Inference Privacy

**Tobias Leemann**[*]
University of Tübingen
Technical University of Munich

**Martin Pawelczyk**[*]
Harvard University

**Gjergji Kasneci**
Technical University of Munich

## Abstract

We propose a novel and practical privacy notion called $f$-Membership Inference Privacy ($f$-MIP), which explicitly considers the capabilities of realistic adversaries under the membership inference attack threat model. Consequently, $f$-MIP offers interpretable privacy guarantees and improved utility (e.g., better classification accuracy). In particular, we derive a parametric family of $f$-MIP guarantees that we refer to as $\mu$-Gaussian Membership Inference Privacy ($\mu$-GMIP) by theoretically analyzing likelihood ratio-based membership inference attacks on stochastic gradient descent (SGD). Our analysis highlights that models trained with standard SGD already offer an elementary level of MIP. Additionally, we show how $f$-MIP can be amplified by adding noise to gradient updates. Our analysis further yields an analytical membership inference attack that offers two distinct advantages over previous approaches. First, unlike existing state-of-the-art attacks that require training hundreds of shadow models, our attack does not require *any* shadow model. Second, our analytical attack enables straightforward auditing of our privacy notion $f$-MIP. Finally, we quantify how various hyperparameters (e.g., batch size, number of model parameters) and specific data characteristics determine an attacker's ability to accurately infer a point's membership in the training set. We demonstrate the effectiveness of our method on models trained on vision and tabular datasets.

## 1 Introduction

Machine learning (ML) has seen a surge in popularity and effectiveness, leading to its widespread application across various domains. However, some of these domains, such as finance and healthcare, deal with sensitive data that cannot be publicly shared due to ethical or regulatory concerns. Therefore, ensuring data privacy becomes crucial at every stage of the ML process, including model development and deployment. In particular, the trained model itself [5, 31] or explanations computed to make the model more interpretable [29, 32] may leak information about the training data if appropriate measures are not taken. For example, this is a problem for recent generative Diffusion Models [7] and Large Language models, where the data leakage seems to be amplified by model size [6].

Differential privacy (DP) [14] is widely acknowledged as the benchmark for ensuring provable privacy in academia and industry [10]. DP utilizes randomized algorithms during training and guarantees that the output of the algorithm will not be significantly influenced by the inclusion or exclusion of any individual sample in the dataset. This provides information-theoretic protection against the maximum amount of information that an attacker can extract about any specific sample in the dataset, even when an attacker has full access to and full knowledge about the predictive model.

While DP is an appealing technique for ensuring privacy, DP's broad theoretical guarantees often come at the expense of a significant loss in utility for many ML algorithms. This utility loss cannot be further reduced by applying savvier algorithms: Recent work [26, 27] confirms that an attacker can

---

[*]Equal contribution. Corresponding authors: `tobias.leemann@uni-tuebingen.de` and `martin.pawelczyk.1@gmail.com`.

37th Conference on Neural Information Processing Systems (NeurIPS 2023).

be implemented whose empirical capacity to differentiate between neighboring datasets $D$ and $D'$ when having access to privatized models matches the theoretical upper bound. This finding suggests that to improve a model's utility, we need to take a step back and inspect the premises underlying DP. For example, previous work has shown that privacy attacks are much weaker when one imposes additional realistic restrictions on the attacker's capabilities [26].

In light of these findings, we revisit the DP threat model and identify three characteristics of an attacker that might be overly restrictive in practice. First, DP grants the attacker full control over the dataset used in training including the capacity to poison all samples in the dataset. For instance, DP's protection includes pathological cases such as an empty dataset and a dataset with a single, adversarial instance [27]. Second, in many applications, it is more likely that the attacker only has access to an API to obtain model predictions [13, 31] or to model gradients [20]. Finally, one may want to protect typical samples from the data distribution. As argued by Triastcyn & Faltings [38], it may be over-constraining to protect images of dogs in a model that is conceived and trained with images of cars.

Such more realistic attackers have been studied in the extensive literature on Membership Inference (MI) attacks (e.g., [5, 41]), where the attacker attempts to determine whether a sample from the data distribution was part of the training dataset. Under the MI threat model, Carlini et al. [5] observe that ML models with very lax ($\epsilon > 5000$) or no ($\epsilon = \infty$) DP-guarantees still provide some defense against membership inference attacks [5, 41]. Hence, we hypothesize that standard ML models trained with low or no noise injection may already offer some level of protection against realistic threats such as MI, despite resulting in very large provable DP bounds.

To build a solid groundwork for our analysis, we present a hypothesis testing interpretation of MI attacks. We then derive $f$-Membership Inference Privacy ($f$-MIP), which bounds the trade-off between an MI attacker's false positive rate (i.e., FPR, type I errors) and false negative rate (i.e., FNR, type II errors) in the hypothesis testing problem by some function $f$. We then analyze the privacy leakage of a gradient update step in stochastic gradient descent (SGD) and derive the first analytically optimal attack based on a likelihood ratio test. However, for $f$-MIP to cover practical scenarios, post-processing and composition operations need to be equipped with tractable privacy guarantees as well. Using $f$-MIP's handy composition properties, we analyze full model training via SGD and derive explicit $f$-MIP guarantees. We further extend our analysis by adding carefully calibrated noise to the SGD updates to show that $f$-MIP may be guaranteed without any noise or with less noise than the same parametric level of $f$-DP [11], leading to a smaller loss of utility.

Our analysis comes with a variety of novel insights: We confirm our hypothesis that, unlike for DP, no noise ($\tau^2 = 0$) needs to be added during SGD to guarantee $f$-MIP. Specifically, we prove that the trade-off curves of a single SGD step converge to the family of Gaussian trade-offs identified by Dong et al. [11] and result in the more specific $\mu$-Gaussian Membership Inference Privacy ($\mu$-GMIP). The main contributions this research offers to the literature on privacy preserving ML include:

1. **Interpretable and practical privacy notion**: We suggest the novel privacy notion of $f$-MIP that addresses the realistic threat of MI attacks. $f$-MIP considers the MI attacker's full trade-off curve between false positives and false negatives. Unlike competing notions, $f$-MIP offers appealing composition and post-processing properties.

2. **Comprehensive theoretical analysis**: We provide (tight) upper bounds on any attacker's ability to run successful MI attacks, i.e., we bound any MI attacker's ability to identify whether points belong to the training set when ML models are trained via gradient updates.

3. **Verification and auditing through novel attacks**: As a side product of our theoretical analysis, which leverages the Neyman-Pearson lemma, we propose a novel set of attacks for auditing privacy leakages. An important advantage of our analytical Gradient Likelihood-Ratio (GLiR) attack is its computational efficiency. Unlike existing attacks that rely on training hundreds of shadow models to approximate the likelihood ratio, our attack does not require any additional training steps.

4. **Privacy amplification through noise addition**: Finally, our analysis shows how one can use noisy SGD (also known as Differentially Private SGD [1]) to reach $f$-MIP while maintaining worst-case DP guarantees. Thereby our work establishes a theoretical connection between $f$-MIP and $f$-DP [11], which allows to translate an $f$-DP guarantee into an $f$-MIP guarantee and vice versa.

## 2 Related Work

**Privacy notions.** DP and its variants provide robust, information-theoretic privacy guarantees by ensuring that the probability distribution of an algorithm's output remains stable even when one sample of the input dataset is changed [14]. For instance, a DP algorithm is $\varepsilon$-DP if the probability of the algorithm outputting a particular subset $E$ for a dataset $S$ is not much higher than the probability of outputting $E$ for a dataset $S_0$ that differs from $S$ in only one element. DP has several appealing features, such as the ability to combine DP algorithms without sacrificing guarantees.

A few recent works have proposed to carefully relax the attacker's capabilities in order to achieve higher utility from private predictions [4, 13, 17, 38]. For example, Dwork & Feldman [13] suggest the notion of "privacy-preserving prediction" to make private model predictions through an API interface. Their work focuses on PAC learning guarantees of any class of Boolean functions. Similarly, Triastcyn & Faltings [38] suggest "Bayesian DP", which is primarily based on the definition of DP, but restricts the points in which the datasets $S$ and $S_0$ may differ to those sampled from the data distribution. In contrast, Izzo et al. [17] introduces a notion based on MI attacks, where their approach guarantees that an adversary $\mathcal{A}$ does not gain a significant advantage in terms of accuracy when distinguishing whether an element $x$ was in the training data set compared to just guessing the most likely option. However, they only constrain the accuracy of the attacker, while we argue that it is essential to bound the entire trade-off curve, particularly in the low FPR regime, to prevent certain re-identification of a few individuals [5]. Our work leverages a hypothesis testing formulation that covers the entire trade-off curve thereby offering protection also to the most vulnerable individuals. Additionally, our privacy notion maintains desirable properties such as composition and privacy amplification through subsampling, which previous notions did not consider.

**Privacy Attacks on ML Models.** Our work is also related to auditing privacy leakages through a common class of attacks called MI attacks. These attacks determine if a given instance is present in the training data of a particular model [5, 7, 9, 15, 23, 29, 30, 31, 32, 35, 36, 40, 41]. Compared to these works, our work suggests a new much stronger class of MI attacks that is analytically derived and uses information from model gradients. An important advantage of our analytically derived attack is its computational efficiency, as it eliminates the need to train any additional shadow models.

## 3 Preliminaries

The classical notion of $(\varepsilon, \delta)$-differential privacy [14] is the current workhorse of private ML and can be described as follows: An algorithm is DP if for any two neighboring datasets $S, S'$ (that differ by one instance) and any subset of possible outputs, the ratio of the probabilities that the algorithm's output lies in the subset for inputs $S, S'$ is bounded by a constant factor. DP is a rigid guarantee, that covers *every* pair of datasets $S$ and $S'$, including pathologically crafted datasets (for instance, Nasr et al. [27] use an empty dataset) that might be unrealistic in practice. For this reason, we consider a different attack model in this work: The MI game [41]. This attack mechanism on ML models follows the goal of inferring an individual's membership in the training set of a learned ML model. We will formulate this problem using the language of hypothesis testing and trade-off functions, a concept from hypothesis testing theory [11]. We will close this section by giving several useful properties of trade-off functions which we leverage in our main theoretical results presented in Sections 4 and 5.

### 3.1 Membership Inference Attacks

The overarching goal of privacy-preserving machine learning lies in protecting personal data. To this end, we will show that an alternative notion of privacy can be defined through the success of a MI attack which attempts to infer whether a given instance was present in the training set or not. Following Yeom et al. [41] we define the standard MI experiment as follows:

**Definition 3.1** (Membership Inference Experiment [41]). *Let $\mathcal{A}$ be an attacker, $A$ be a learning algorithm, $N$ be a positive integer, and $\mathcal{D}$ be a distribution over data points $x \in D$, where the vector $x$ may also be a tuple of data and labels. The MI experiment proceeds as follows: The model and data owner $\mathcal{O}$ samples $S \sim \mathcal{D}_N$ (i.e, sample n points i.i.d. from $\mathcal{D}$) and trains $A_S = A(S)$. They choose $b \in \{0, 1\}$ uniformly at random and draw $x' \sim \mathcal{D}$ if $b = 0$, or $x' \sim S$ if $b = 1$. Finally, the attacker is successful if $\mathcal{A}(x', A_S, N, \mathcal{D}) = b$. $\mathcal{A}$ must output either 0 or 1.*

We note that the membership inference threat model features several key differences to the threat model underlying DP, which are listed in Table 1. Most notably, in MI attacks, the datasets are sampled from the distribution $\mathcal{D}$, whereas DP protects all datasets. This corresponds to granting the attacker the capacity of full dataset manipulation. Therefore, the MI attack threat model is sensible in cases where the attacker cannot manipulate the dataset through injection of malicious samples also called "canaries". This may be realistic for financial and healthcare applications, where the data is often collected from actual events (e.g., past trades) or only a handful of people (trusted hospital staff) have access to the records. In such scenarios, it might be overly restrictive to protect against worst-case canary attacks as attackers cannot freely inject arbitrary records into the training datasets. Furthermore, MI attacks are handy as a fundamental ingredient in crafting data extraction attacks [6]. Hence we expect a privacy notion based on the MI threat model to offer protection against a broader class of reconstruction attacks. Finally, being an established threat in the literature [5, 9, 31, 40, 41], MI can be audited through a variety of existing attacks.

|  | f-DP threat model | f-MIP threat model (this work) |
|---|---|---|
| Goal | Distinguish between $S$ and $S'$ for *any* $S, S'$ that differ in at most one instance. | Distinguish whether $\boldsymbol{x}' \in S$ (training data set) or not. |
| Dataset access | Attacker has full data access. For example, the attacker can poison or adversarially construct datasets on which ML models could be trained; e.g., $S = \{\}$ and $S' = \{10^6\}$. | Attacker has no access to the training data set; i.e., the model owner privately trains their model free of adversarially poisoned samples. |
| Protected Instances | The instance in which $S$ and $S'$ differ is arbitrary. This includes OOD samples and extreme outliers. | The sample $\boldsymbol{x}'$ for which membership is to be inferred is drawn from the data distribution $\mathcal{D}$. Therefore, MI is concerned with typical samples that can occur in practice. |
| Best used | When the specific attack model is unknown. Offers a form of general protection. | When dataset access (e.g. canary injection) of an attacker can be ruled out and the main attack goal lies in revealing private training data (e.g., membership inference, data reconstruction). |
| Model knowledge | The attacker knows the model architecture and has full access to the model in form of its parameters, hyperparameters and its model outputs. | |

Table 1: Comparing the threat models underlying $f$-DP and $f$-MIP.

## 3.2 Membership Inference Privacy as a Hypothesis Testing Problem

While DP has been studied through the perspective of a hypothesis testing formulation for a while [3, 11, 19, 39], we adapt this route to formulate membership inference attacks. To this end, consider the following hypothesis test:

$$\mathrm{H}_0 : \boldsymbol{x}' \notin S \text{ vs. } \mathrm{H}_1 : \boldsymbol{x}' \in S. \tag{1}$$

Rejecting the null hypothesis corresponds to detecting the presence of the individual $\boldsymbol{x}'$ in $S$, whereas failing to reject the null hypothesis means inferring that $\boldsymbol{x}'$ was not part of the dataset $S$. The formulation in (1) is a natural vehicle to think about any attacker's capabilities in detecting members of a train set in terms of false positive and true positive rates. The motivation behind these measures is that the attacker wants to reliably identify the subset of data points belonging to the training set (i.e., true positives) while incurring as few false positive errors as possible [5]. In other words, the attacker wants to maximize their true positive rate at any chosen and ideally low false positive rate (e.g., 0.001). From this perspective, the formulation in (1) allows to define membership inference privacy via trade-off functions $f$ which exactly characterize the relation of false negative rates (i.e., 1-TPR) and false positive rates that an optimal attacker can achieve.

**Definition 3.2.** *(Trade-off function [11]) For any two probability distributions $P$ and $Q$ on the same space, denote the trade-off function $Test(P; Q) : [0; 1] \to [0; 1]$*

$$\mathrm{Test}(P; Q)(\alpha) = \inf \{FNR \mid FPR = \alpha\}, \tag{2}$$

*where the infimum is taken over all (measurable) rejection rules ("tests") which lead to a FPR of $\alpha$ between distributions $P, Q$.*

Not every function makes for a valid trade-off function. Instead, trade-off functions possess certain characteristics that are handy in their analysis.

**Definition 3.3** (Characterization of trade-off functions [11]). *A function $f : [0,1] \to [0,1]$ is a trade-off function if $f$ is convex, continuous at zero, non-increasing, and $f(r) \leq 1 - r$ for $r \in [0,1]$.*

We additionally introduce a semi-order on the space of trade-off functions to make statements on the hardness of different trade-offs in relation to each other.

**Definition 3.4** (Comparing trade-offs). *A trade-off function $f$ is uniformly at least as hard as another trade-off function $g$, if $f(r) \geq g(r)$ for all $0 \leq r \leq 1$. We write $f \geq g$.*

If $\text{Test}(P;Q) \geq \text{Test}(P';Q')$, testing $P$ vs $Q$ is uniformly at least as hard as testing $P'$ vs $Q'$. Intuitively, this means that for a given FPR $\alpha$, the best test possible test on $(P;Q)$ will result in an equal or higher FNR than the best test on $(P';Q')$.

### 3.3 Noisy Stochastic Gradient Descent (Noisy SGD)

Most recent large-scale ML models are trained via stochastic gradient descent (SGD). Noisy SGD (also known as DP-SGD) is a variant of classical SGD that comes with privacy guarantees. We consider the algorithm as in the work by Abadi et al. [1], which we restate for convenience in Appendix A. While its characteristics with respect to DP have been extensively studied, we take a fundamentally different perspective in this work and study the capabilities of this algorithm to protect against membership inference attacks. In summary, the algorithm consists of three fundamental steps: *gradient clipping* (i.e., $\boldsymbol{\theta}_i := \boldsymbol{g}(\boldsymbol{x}_i, y_i) \cdot \max(1, C/\|\boldsymbol{g}(\boldsymbol{x}_i, y_i)\|)$ where $\boldsymbol{g}(\boldsymbol{x}_i, y_i) = \nabla \mathcal{L}(\boldsymbol{x}_i, y_i)$ is the gradient with respect to the loss function $\mathcal{L}$), *aggregation* (i.e., $\boldsymbol{m} = \frac{1}{n} \sum_{i=1}^{n} \boldsymbol{\theta}_i$) and *adding Gaussian noise* (i.e., $\tilde{\boldsymbol{m}} = \boldsymbol{m} + Y$ where $Y \sim \mathcal{N}(\boldsymbol{0}, \tau^2 \boldsymbol{I})$ with variance parameter $\tau^2$). To obtain privacy bounds for this algorithm, we study MI attacks for means of random variables. This allows us to bound the MI vulnerability of SGD.

## 4 Navigating Between Membership Inference Privacy and DP

In this section, we formally define our privacy notion $f$-MIP. To this end, it will be handy to view MI attacks as hypothesis tests.

### 4.1 Membership Inference Attacks from a Hypothesis Testing Perspective

Initially, we define the following distributions of the algorithm's output

$$A_0 = A(\boldsymbol{X} \cup \{\boldsymbol{x}\}) \text{ with } \boldsymbol{X} \sim \mathcal{D}^{n-1}, \boldsymbol{x} \sim \mathcal{D} \text{ and } A_1(\boldsymbol{x}') = A(\boldsymbol{X} \cup \{\boldsymbol{x}'\}) \text{ with } \boldsymbol{X} \sim \mathcal{D}^{n-1}, \quad (3)$$

where we denote other randomly sampled instances that go into the algorithm by $\boldsymbol{X} = \{\boldsymbol{x}_1, ... \boldsymbol{x}_{n-1}\}$. Here $A_0$ represents the output distribution under the null hypothesis ($H_0$) where the sample $\boldsymbol{x}'$ is not part of the training dataset. On the other hand, $A_1$ is the output distribution under the alternative hypothesis ($H_1$) where $\boldsymbol{x}'$ was part of the training dataset. The output contains randomness due to the instances drawn from the distribution $\mathcal{D}$ and due to potential inherent randomness in $A$.

We observe that the distribution $A_1$ depends on the sample $\boldsymbol{x}'$ which is known to the attacker. The attacker will have access to samples for which $A_0$ and $A_1(\boldsymbol{x}')$ are simpler to distinguish and others where the distinction is harder. To reason about the characteristics of such a stochastically composed test, we define a composition operator that defines an optimal test in such a setup. To obtain a global FPR of $\alpha$, an attacker can target different FPRs $\bar{\alpha}(\boldsymbol{x}')$ for each specific test. We need to consider the optimum over all possible ways of choosing $\bar{\alpha}(\boldsymbol{x}')$, which we refer to as *test-specific FPR function*, giving rise to the following definition.

**Definition 4.1** (Stochastic composition of trade-off functions). *Let $\mathcal{F}$ be a family of trade-off functions, $h : D \subset \mathbb{R}^d \to \mathcal{F}$ be a function that maps an instance of the data domain to a corresponding trade-off function, and $\mathcal{D}$ be a probability distribution on $D$. The set of valid test-specific FPR functions $\bar{\alpha} : D \to [0,1]$ that result in a global FPR of $\alpha \in [0,1]$ given the distribution $\mathcal{D}$ is defined through*

$$\mathcal{E}(\alpha, \mathcal{D}) = \{\bar{\alpha} : D \to [0,1] \mid \mathbb{E}_{\boldsymbol{x}' \sim \mathcal{D}} [\bar{\alpha}(\boldsymbol{x}')] = \alpha\}. \quad (4)$$

*For a given test-specific FPR function, $\bar{\alpha}$ the global false negative rate (type II error) $\beta$ is given by*

$$\beta_h(\bar{\alpha}) = \mathbb{E}_{\boldsymbol{x}' \sim \mathcal{D}} \left[ h(\boldsymbol{x})(\bar{\alpha}(\boldsymbol{x})) \right], \tag{5}$$

*where $\bar{\alpha}(\boldsymbol{x})$ is the argument to the trade-off function $h(\boldsymbol{x}) \in \mathcal{F}$. For a global $\alpha \in [0,1]$ the stochastic composition of these trade-functions is defined as*

$$\left( \bigotimes_{\boldsymbol{x} \sim \mathcal{D}} h(\boldsymbol{x}) \right) (\alpha) = \min_{\bar{\alpha} \in \mathcal{E}(\alpha, \mathcal{D})} \left\{ \beta_h(\bar{\alpha}) \right\}, \tag{6}$$

*(supposing the minimum exists), the smallest global false negative rate possible at a global FPR of $\alpha$.*

This definition specifies the trade-off function $\bigotimes_{\boldsymbol{x} \sim \mathcal{D}} h(\boldsymbol{x}) : [0,1] \to [0,1]$ of a stochastic composition of several trade-offs. While it is reminiscent of the "most powerful test" (MPT) [28], there are several differences to the MPT that are important in our work. Most prominently, a straightforward construction of the MPT to MI problems does not work since the adversary does not only run one hypothesis test to guess whether one sample belongs to the training data set or not; instead, the adversary draws multiple samples and runs sample-dependent and (potentially) different hypotheses tests for each drawn sample. This is necessary due to the form of the alternative hypotheses in the formulation of the test in (3), which depends on the sample $\boldsymbol{x}'$. We therefore require a tool to compose the results from different hypothesis tests. Finally, we prove that the trade-off of the stochastic composition has the properties of a trade-off function (see App. D.1):

**Theorem 4.1** (Stochastic composition of trade-off functions). *The stochastic composition $\bigotimes_{\boldsymbol{x} \sim \mathcal{D}} h(\boldsymbol{x})$ of trade-off functions $h(\boldsymbol{x})$ maintains the characteristics of a trade-off function, i.e., it is convex, non-increasing, $\left( \bigotimes_{\boldsymbol{x} \sim \mathcal{D}} h(\boldsymbol{x}) \right)(r) \leq 1 - r$ for all $r \in [0,1]$, and it is continuous at zero.*

### 4.2 $f$-Membership Inference Privacy ($f$-MIP)

This rigorous definition of the stochastic composition operator allows us to define membership inference privacy from a hypothesis testing perspective.

**Definition 4.2** ($f$-Membership Inference Privacy). *Let $f$ be a trade-off function. An algorithm[1] $A : D^n \to \mathbb{R}^d$ is said to be $f$-membership inference private ($f$-MIP) with respect to a data distribution $\mathcal{D}$ if*

$$\bigotimes_{\boldsymbol{x}' \sim \mathcal{D}} \text{Test} \left( A_0; A_1(\boldsymbol{x}') \right) \geq f, \tag{7}$$

*where $\boldsymbol{x}' \sim \mathcal{D}$ and $\bigotimes$ denotes the stochastic composition built from individual trade-off functions of the MI hypotheses tests for random draws of $\boldsymbol{x}'$.*

In this definition, both sides are functions dependent on the false positive rate $\alpha$. A prominent special case of a trade-off function is the Gaussian trade-off, which stems from testing one-dimensional normal distributions of unit variance that are spaced apart by $\mu \in \mathbb{R}_{\geq 0}$. Therefore, defining the following special case of $f$-MIP will be useful.

**Definition 4.3** ($\mu$-Gaussian Membership Inference Privacy). *Let $\Phi$ be the cumulative distribution function (CDF) of a standard normal distribution. Define $g_\mu(\alpha) := \Phi(\Phi^{-1}(1-\alpha) - \mu)$ to be the trade-off function derived from testing two Gaussians; one with mean $0$ and one with mean $\mu$. An algorithm $A$ is $\mu$-Gaussian Membership Inference private ($\mu$-GMIP) with privacy parameter $\mu$ if it is $g_\mu$-MIP, i.e., it is MI private with trade-off function $g_\mu$.*

**Remark 4.1.** *DP can also be defined via the Gaussian trade-off function, which results in $\mu$-Gaussian Differential Privacy ($\mu$-GDP, [11]). While the trade-off curves for both $\mu$-GDP and $\mu$-GMIP have the same parametric form, they have different interpretations: $\mu$-GDP describes the trade-off function that an attacker with complete knowledge (left column in Table 1) could achieve while $\mu$-GMIP describes the trade-off function that an attacker with MI attack capability can achieve (right column in Table 1). In the next section we will quantify their connection further.*

---

[1] When using the term "algorithm", we also include randomized mappings.

### 4.3 Relating $f$-MIP and $f$-DP

We close this section by providing first results regarding the relation between $f$-DP and $f$-MIP. As expected, $f$-DP is strictly stronger than $f$-MIP, which can be condensed in the following result:

**Theorem 4.2** ($f$-DP implies $f$-MIP). *Let an algorithm $A : D^n \to \mathbb{R}^d$ be $f$-differentially private [11]. Then, algorithm $A$ will also be $f$-membership inference private.*

We proof this result in Appendix F.1. This theorem suggests one intuitive, simple and yet actionable approach to guarantee Membership Inference Privacy. This approach involves the use of DP learning algorithms such as DP-SGD [1], which train models using noised gradients. However, as we will see in the next section, using noise levels to guarantee $f$-DP is usually suboptimal to guarantee $f$-MIP.

## 5 Implementing $f$-MIP through Noisy SGD

We would now like to obtain a practical learning algorithm that comes with $f$-MIP guarantees. As the dependency between the final model parameters and the input data is usually hard to characterize, we follow the common approach and trace the information flow from the data to the model parameters through the training process of stochastic gradient descent [1, 33]. Since the gradient updates are the only path where information flows from the data into the model, it suffices to privatize this step.

### 5.1 $f$-MIP for One Step of Noisy SGD

We start by considering a single SGD step. Following prior work [1, 33], we make the standard assumption that only the mean over the individual gradients $m = \frac{1}{n} \sum_{i=1}^{n} \boldsymbol{\theta}_i$ is used to update the model (or is published directly) where $\boldsymbol{\theta}_i \in \mathbb{R}^d$ is a sample gradient. Consistent with the definition of the membership inference game, the attacker tries to predict whether a specific gradient $\boldsymbol{\theta}'$ was part of the set $\{\boldsymbol{\theta}_i\}_i$ that was used to compute the model's mean gradient $m$ or not. We are interested in determining the shape of the attacker's trade-off function. For the sake of conciseness, we directly consider one step of noisy SGD (i.e., one averaging operation with additional noising, see Algorithm 2 from the Appendix), which subsumes a result for the case without noise by setting $\tau^2 = 0$. We establish the following theorem using the Central Limit Theorem (CLT) for means of adequately large batches of $n$ sample gradients, which is proven in Appendix E.

**Theorem 5.1** (One-step noisy SGD is $f$-membership inference private). *Denote the cumulative distribution function of the non-central chi-squared distribution with $d$ degrees of freedom and non-centrality parameter $\gamma$ by $F_{\chi_d^2(\gamma)}$. Let the gradients $\boldsymbol{\theta}' \in \mathbb{R}^d$ of the test points follow a distribution with mean $\boldsymbol{\mu}$ and covariance $\boldsymbol{\Sigma}$, let $K \geq \|\boldsymbol{\Sigma}^{-1/2}\boldsymbol{\theta}'\|_2^2$ and define $n_{effective} = n + \frac{\tau^2 n^2}{C^2}$. For sufficiently large batch sizes $n$, one step of noisy SGD is $f$-membership inference private with trade-off function given by:*

$$\beta(\alpha) \approx 1 - F_{\chi_d^2((n_{effective}-1)K)} \left( \frac{n_{effective}}{n_{effective} - 1} F^{-1}_{\chi_d^2(n_{effective}K)}(\alpha) \right). \tag{8}$$

The larger the number of parameters $d$ and the batch size $n$ grow, the more the trade-off curve approaches the $\mu$-GMIP curve, which we show next (see Figure 1).

**Corollary 5.1** (One step noisy SGD is approx. $\mu$-GMIP). *For large $d, n$, noisy SGD is approximately $g_{\mu_{Step}}$-GMIP. In particular, $\beta(\alpha) \approx \Phi(\Phi^{-1}(1 - \alpha) - \mu_{Step})$ with privacy parameter:*

$$\mu_{step} = \frac{d + (2n_{effective} - 1)K}{n_{effective}\sqrt{2d + 4n_{effective}K}}. \tag{9}$$

This result is striking in its generality as it also covers models trained without additional noise or gradient cropping ($n_{\text{effective}} = n$ in that case). Unlike for DP, even standard models trained with non-noisy SGD offer an elementary level of MIP. Our result further explicitly quantifies four factors

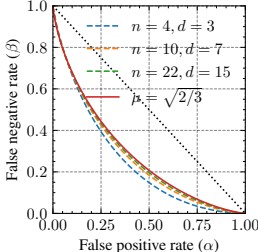

Figure 1: **Trade-off function convergence.** The trade-off function from Theorem 5.1 converges to the one from Corollary 5.1 where $\tau^2{=}0$ and $K{=}d$.

that lead to attack success: the batch size $n$, the number of parameters $d$, the strength of the noise $\tau^2$ and the worst-case data-dependent *gradient susceptibility* $\|\mathbf{\Sigma}^{-1/2}\boldsymbol{\theta}'\|_2^2$. The closeness of the trade-off function to the diagonal, which is equivalent to the attacker randomly guessing whether a gradient $\boldsymbol{\theta}'$ was part of the training data or not, is majorly determined by the ratio of $d$ to $n$. The higher the value of $d$ relative to $n$, the easier it becomes for the attacker to identify training data points. Furthermore, a higher gradient susceptibility $K$, which measures the atypicality of a gradient with respect to the gradient distribution, increases the likelihood of MI attacks succeeding in identifying training data membership. It is worth noting that if we do not restrict the gradient distribution or its support, then there might always exist gradient samples that significantly distort the mean, revealing their membership in the training dataset. This phenomenon is akin to the $\delta$ parameter in DP, which also allows exceptions for highly improbable events.

**Remark 5.1** (Magnitude of $\mu_{\text{Step}}$). *When the dimensions of the uncorrelated components in $\mathbf{\Sigma}^{-\frac{1}{2}}\boldsymbol{\theta}'$ are also independent, we expect $K$ to follow a $\chi^2$-distribution with $d$ degrees of freedom and thus $K \in \mathcal{O}(d)$. In the standard SGD-regime ($\tau^2 = 0$) with $d, n \gg 1$, we obtain $\mu \in \mathcal{O}\big(\sqrt{d/n}\big)$.*

**Remark 5.2** (On Optimality). *The dependency on $d$ when $\tau^2 > 0$ is a consequence of our intentionally broad proving strategy. Our proof approach consists of two key steps: First, we establish the optimal LRT under general gradient distributions, without adding noise or imposing any cropping constraints (See Appendix E.1). This initial step serves as the foundation for our subsequent analysis and is (1) as general as possible covering all distributions with finite variance and is (2) optimal in the sense of the Neyman-Pearson Lemma, i.e., it cannot be improved. This means that our result covers all models trained with standard SGD ($\tau^2 = 0$ and $C = \infty$) and is remarkable in its generality as it is the first to suggest clear conditions when adding noise is not required to reach $f$-MIP. Second, we specialize our findings to cropped random variables with added noise (See Appendix E.2). This analysis could potentially be improved by considering individual gradient dimensions independently.*

### 5.2 Composition and Subsampling

In the previous section, we have derived the trade-off function for a single step of SGD. Since SGD is run over multiple rounds, we require an understanding of how the individual trade-off functions can be composed when a sequence of $f$-MIP operations is conducted, and a random subset of the entire data distribution is used as an input for the privatized algorithm. The next lemma provides such a result for $\mu$-GMIP and follows from a result that holds for hypotheses tests between Gaussian random variables due to Dong et al. [11] (see Appendix D.3 for details and more results).

**Lemma 5.1** (Asymptotic convergence of infinite DP-SGD). *Let $n$ be the batch size in SGD, and $N$ be the entire size of the dataset. If a single SGD-Step is at least as hard as $\mu_{step}$-GMIP with respect to the samples that were part of the batch and $\frac{n\sqrt{t}}{N} \to c$ as $\lim_{t \to \infty}$ (the batch size is gradually decreased), then the noisy SGD algorithm will be $\mu$-GMIP with*

$$\mu = \sqrt{2}c\sqrt{\exp(\mu_{step}^2)\Phi\left(1.5\mu_{step}\right) + 3\Phi\left(-0.5\mu_{step}\right) - 2}. \tag{10}$$

Note that this result also provides a (loose) bound for the case where exactly $T$ iterations are run with a batch size of $n'$ with $c = \frac{n'\sqrt{T}}{N}$ (through using $n(t) = n'$ if $t \le T$, else $n(t) = \frac{n'\sqrt{T}}{\sqrt{t}}$). With this result in place, we can defend against MI attacks using the standard noisy SGD algorithm.

## 6 Experimental Evaluation

**Datasets and Models.** We use three datasets that were previously used in works on privacy risks of ML models [32]: The CIFAR-10 dataset which consists of 60k small images [21], the Purchase tabular classification dataset [25] and the Adult income classification dataset from the UCI machine learning repository [12]. Following prior work by Abadi et al. [1], we use a model pretrained on CIFAR-100 and finetune the last layer on CIFAR-10 using a ResNet-56 model for this task [16] where the number of fine-tuned parameters equals $d = 650$. We follow a similar strategy on the Purchase dataset, where we use a three-layer neural network. For finetuning, we use the 20 most common classes and $d = 2580$ parameters while the model is pretrained on 80 classes. On the adult dataset, we use a two-layer network with 512 random features in the first layer trained from scratch on the

dataset such that $d = 1026$. We refer to Appendix C.1 for additional training details. We release our code online.[2]

## 6.1 Gradient Attacks Based on the Analytical LRT

To confirm our theoretical analysis for one step of SGD and its composition, we implement the gradient attack based on the likelihood ratio test derived in the proof of Theorem 5.1. We provide a sketch of the implementation in Algorithm 1 and additional details in Appendix C.3. An essential requirement in the construction of the empirical test is the estimation of the true gradient mean $\boldsymbol{\mu}$ and the true inverse covariance matrix $\boldsymbol{\Sigma}^{-1}$ since these quantities are essential parts of both the test statistic $\hat{S}$ and the true gradient susceptibility term $\hat{K}$ needed for the analytical attack. The attacker uses

---

**Algorithm 1: Gradient Likelihood Ratio (GLiR) Attack**

**Require:** Training data distribution $\mathcal{D}$, batch size $n$, number of parameters $d$, query point $\boldsymbol{x} \in D$, averaged gradients of each batch $\boldsymbol{m}_t \in \mathbb{R}^d$ for training steps $t = 1, \ldots, T$, parameter gradient computation function $\nabla_{\boldsymbol{w}_t}\mathcal{L} : D \to \mathbb{R}^d$ of training, threshold $\eta$

1: $p_{\text{total}} \leftarrow 0$
2: **for** $t = 1, \ldots, T$ **do**
3:     $B = \{\boldsymbol{b}_1, \ldots, \boldsymbol{b}_m\} \sim D^m$     ▷ Sample background data
4:     $\boldsymbol{g}_i = \nabla_{\boldsymbol{w}}\mathcal{L}(\boldsymbol{b}_i), i = 1...m$   ▷ Compute background gradients
5:     $\hat{\boldsymbol{\Sigma}} = \text{Cov}\{\boldsymbol{g}_1, ..., \boldsymbol{g}_m\} \in \mathbb{R}^{d \times d}$  ▷ Approximate covariance $\boldsymbol{\Sigma}$
6:     $\hat{\boldsymbol{\mu}} = \text{Mean}\{\boldsymbol{g}_1, ..., \boldsymbol{g}_m\} \in \mathbb{R}^d$      ▷ Approximate mean $\boldsymbol{\mu}$
7:     $\boldsymbol{\theta} = \nabla_{w_t}\mathcal{L}(\boldsymbol{x})$    ▷ Compute gradients for the query point
8:     $\hat{S} = (n-1)(\boldsymbol{m}_t - \boldsymbol{\theta})^\top \hat{\boldsymbol{\Sigma}}^{-1}(\boldsymbol{m}_t - \boldsymbol{\theta})$ ▷ Compute test statistic
9:     $\hat{K} = \|\hat{\boldsymbol{\Sigma}}^{-1/2}(\boldsymbol{\theta} - \hat{\boldsymbol{\mu}})\|_2^2$    ▷ Estimate gradient susceptibility
10:    $p_{\text{step}} = \log \text{F}^{-1}_{\chi^2_d(n\hat{K})}(\hat{S})$   ▷ Compute $\log p$-value under $H_0$
11:    $p_{\text{total}} \leftarrow p_{\text{total}} + p_{\text{step}}$
12: **end for**
13: **return** Train if $p_{\text{total}} < \eta$, **else** Test

---

their access to the gradient distribution (which is standard for membership inference attacks [5, 29] and realistic in federated learning scenarios [20]), to estimate the distribution parameters. In practice, however, the empirical estimates of $\hat{\boldsymbol{\mu}}, \hat{\boldsymbol{\Sigma}}^{-1}$ and thus $\hat{K}$ will be noisy and therefore we do not expect that the empirical trade-off curves match the analytical curves exactly.

Using our novel Gradient Likelihood Ratio (GLiR) attack we can audit our derived guarantees and their utility. First, we audit our one-step guarantees from Theorem 5.1. To compare the models, we adapt the batch size $n$ such that all models reach the same level of $\mu$-GMIP. In Figure 2a, we use a simulated gradient distribution with known parameters $\boldsymbol{\mu}, \boldsymbol{\Sigma}^{-1}$ and $d$. In this case, we can estimate $K$ accurately and observe that our bounds are tight when the distribution parameters and thus the respective gradient susceptibilities can be computed accurately. We provide additional ablation studies that gauge the approximation quality of with small values for $d, n$ and different simulated distributions in Appendix C.2. When the parameters are unknown and we have to estimate the parameters, our attacks become weaker and do not match the analytical prediction (see Figure 2b).

We also audit our composition guarantees. We do five SGD-steps in Figure 2c. While there is a small gain in attack performance on the CIFAR-10 dataset (e.g., at FPR=0.25), the attack performance on the other datasets remains largely unaffected. This mismatch occurs since the theoretical analysis is based on the premise that the attacker gains access to independently sampled gradient means for each step to separate training and non-training points, but in practice we do not gain much new information as the model updates are not statistically independent and too incremental to change the gradient means significantly between two subsequent steps. Therefore, a practical attacker does not gain much additional information through performing several steps instead of one. Future work is required to model these dependencies and potentially arrive at a tighter composition result under incremental parameter updates. We provide results for additional existing membership inference attacks, for instance the recent loss-based likelihood-ratio attack by Carlini et al. [5] in Appendix C.4, which all show weaker success rates than the gradient-based attack that proved most powerful in our setting.

## 6.2 Comparing Model Utility under $\mu$-GDP and $\mu$-GMIP

Here we compare the utility under our privacy notion to the utility under differential privacy. We sample 20 different privacy levels ranging from $\mu \in [0.4, ..., 50]$ and calibrate the noise in the SGD iteration to reach the desired value of $\mu$. We can do so both for $\mu$-GMIP using the result in Equation (10) and using the result by Dong et al. [11, Corollary 4] for $\mu$-GDP, which result in the same

---

[2]https://github.com/tleemann/gaussian_mip

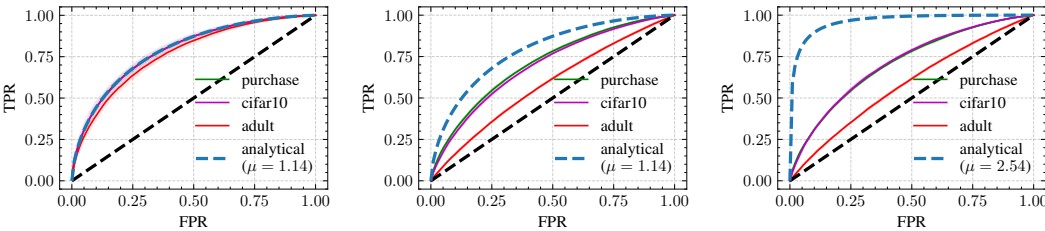

(a) Single step of simulated gradient distribution with known parameters.

(b) Single step with real model gradients and estimated parameters.

(c) As in (b), but now composition of 5 steps for real model gradients.

Figure 2: **Auditing $f$-MIP with our gradient attack (GLiR) when $\tau^2 = 0$.** We show trade-off curves when the gradient distribution is known (a) and when the gradients are obtained from a trained model that was finetuned on various data sets (b, c). The analytical solutions are computed with a value of $K = d$ and using the composition result for $k$ steps in Appendix D.3 for (c).

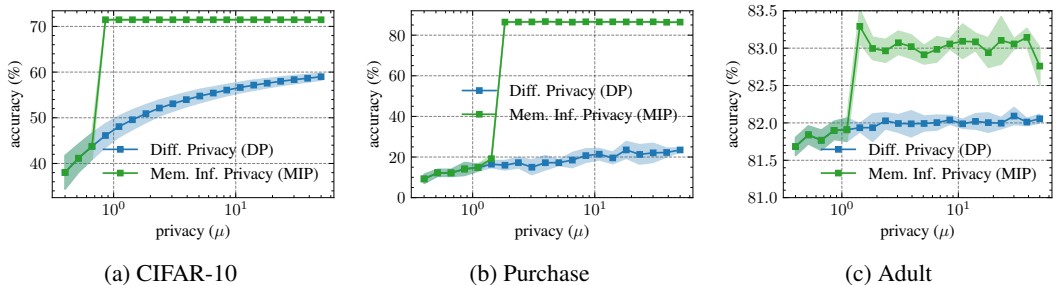

(a) CIFAR-10

(b) Purchase

(c) Adult

Figure 3: **Utility of DP versus MIP.** Model performance on three datasets across different privacy levels $\mu$ (small $\mu$ denotes high privacy) using the notions of $\mu$-Gaussian Differential Privacy (parametric form of $f$-DP, [11]) and $\mu$-Gaussian Membership Inference Privacy (parametric form of $f$-MIP, ours) on three datasets. GMIP usually allows for substantially increased accuracy over the corresponding GDP guarantee with the same attack success rates controlled by $\mu$. However, the attacker under GMIP runs membership inference (MI) attacks while GDP allows for a wider set of privacy threat models. For more details on differences in the underlying threat models see Table 1.

attack success rates while $\mu$-GDP allows for stronger privacy threat models. Due to Theorem 4.2, we never need to add more noise for $\mu$-GMIP than for $\mu$-DP. Further details are provided in Appendix C.1. Figure 3 shows a comparison of the accuracy that the models obtain. We observe that the model under GMIP results in significantly higher accuracy for most values of $\mu$. As $\mu \to 0$ both privacy notions require excessive amounts of noise such that the utility decreases towards the random guessing accuracy. On the other hand, for higher values of $\mu$, there is no need to add any noise to the gradient to obtain $\mu$-GMIP, allowing to obtain the full utility of the unconstrained model. This indicates that useful GMIP-bounds do not necessarily require noise. For instance, on the CIFAR-10 model, no noise is required for $\mu \geq 0.86$ which is a reasonable privacy level [11]. Overall, these results highlight that useful and interpretable privacy guarantees can often be obtained without sacrificing utility.

## 7  Conclusion and Future Work

In the present work, we derived the general notion of $f$-Membership Inference Privacy ($f$-MIP) by taking a hypothesis testing perspective on membership inference attacks. We then studied the noisy SGD algorithm as a model-agnostic tool to implement $f$-Membership Inference Privacy, while maintaining Differential Privacy (DP) as a worst-case guarantee. Our analysis revealed that significantly less noise may be required to obtain $f$-MIP compared to DP resulting in increased utility. Future work is required to better model the dependencies when composing subsequent SGD steps which could lead to improved bounds in practice. Furthermore, our analysis shows that when the capacity of the attacker is further restricted, e.g., to API access of predictions, there remains a gap between our theoretical bounds and loss-based membership inference attacks that can be implemented for real models. More work is required to either produce more sophisticated attacks or derive theoretical bounds for even less powerful attackers to close this gap.

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
