# OpenReview forum: "Gaussian Membership Inference Privacy"
_NeurIPS.cc/2023/Conference — NeurIPS 2023 poster_

### Official Review · Reviewer_vdUQ · 2023-07-04

**Soundness:** 3 good
**Presentation:** 3 good
**Contribution:** 3 good
**Rating:** 5
**Confidence:** 3

**Summary:**

The proposed f-MIP method incorporates a practical membership inference attack threat model, offering easily interpretable privacy guarantees. This approach improves utility, especially when the attacker's capabilities are realistically constrained. Through a theoretical analysis of likelihood ratio-based membership inference attacks on noisy stochastic gradient descent (SGD), μ-Gaussian Membership Inference Privacy (μ-GMIP) is introduced. μ-GMIP requires less noise compared to the corresponding Gaussian differential privacy (GDP) guarantees, resulting in higher utility.

**Strengths:**

-	The concept of applying control over type I and type II errors to MIP is interesting.
-	The proposed approach holds promise for enhancing privacy protection in SGD.
-	The numerical study provides evidence that the proposed method outperforms noisy SGD in terms of utility, highlighting its potential for practical applications.


**Weaknesses:**

-	The paper lacks a theoretical exploration of the relationship between f-MIP and f-DP, which hinders a comprehensive understanding of the proposed method. For instance, while mu-GMIP and mu-GDP are compared at the same mu in Figure 1, it remains unclear whether they actually share the same mu value.
-	The absence of a discussion on post-processing or the composition rule (in terms of mu) limits the practical applicability and usefulness of the proposed approach.


**Questions:**

As mentioned in the paper, it is necessary to focus on cases where the membership is accurately determined among those classified as members, which corresponds to achieving a high true positive rate (TPR) at low false positive rate (FPR) in threshold-type tests. Therefore, it would be logical to concentrate on scenarios with low FPR. However, the trade-off function is compared across the entire range [0, 1] rather than solely focusing on these low FPR cases.

**Limitations:**

-	The iid assumption is not realistic in practical scenarios, which suggests that the actual privacy guarantees would be lower than those stated in the paper.
-	Since the f-MIP method is specifically designed for SGD, it raises questions about its generalizability to other problem domains.

---

> ### Author Rebuttal · Authors · 2023-08-07
>
> We thank the reviewer for their review and for referring to our approach as interesting and promising. We will answer the specific questions below.
>
> > The paper lacks a theoretical exploration of the relationship between f-MIP and f-DP [...]  in Figure 1, it remains unclear whether they actually share the same mu value.
>
> The value of $\mu$ defines the possible trade-off curve attainable by an attacker. This can be seen as a proxy for susceptibility of a model to specific types of attack. Regarding Figure 1, we compute noise levels $\tau$, such that the privacy notions share the same nominal value of $\mu$. The required noise levels across both f-DP and f_MIP are different and shown in Table 4 (App.), where we show that more noise is required for $\mu$-GDP than for $\mu$-GMIP.  The premises underlying MIP and DP are different however. We refer to the general response where we explicitly compare the threat models more clearly.
>
> The relationship between the $\mu$ values from G-MIP and G-DP can also be made explicit through the following Corollary:
>
> **Corollary: Converting Trade-Offs between Gaussian-DP and Gaussian-MIP**
>
> Under the usual setting of $K=d$ and the conditions stated in Corollary 4.1., the following conversions between the privacy parameter $\mu$ of a single step of noisy SGD hold:
>
> 1. Converting $\mu_\text{DP}$ into $\mu_\text{MIP}$
>
> $\mu_{\text{MIP}} = \sqrt{\frac{d}{n +\frac{4C^2}{\mu_{\text{DP}}^2}+\frac{1}{2}}}$
>
> 2. Converting $\mu_\text{MIP}$ into $\mu_\text{DP}$
>
> $\mu_{\text{DP}} = \begin{cases}
>     \frac{2}{\sqrt{\frac{d}{\mu_{\text{MIP}}^2}-n -\frac{1}{2}}}, & \text{if } \mu_{\text{MIP}} < \sqrt{\frac{2d}{2n+1}} \\\\
>    \infty, & \text{else}
>   \end{cases}$
>
> **Proof Sketch.** This result follows from solving Corollary 4.2 and the result from Dong et al. [1, last line on page 30] for differential privacy of a single SGD step, $\mu_{\text{DP}} = \frac{1}{\sigma} = \frac{2C}{\tau n}$ (where Dong et al.'s $\sigma= \frac{\tau n}{2C}$ in our notation) for $\tau$, setting the two terms equal, and solving for $\mu_{\text{MIP}}$ or $\mu_{DP}$ correctly. The same strategy can be applied for composed results.
>
> We observe that, as expected, $\mu_{\text{DP}}$ and $\mu_{\text{MIP}}$ increase and decrease mutually. For high values of $\mu_{\text{MIP}}$ (weak privacy without any additional noise added), no DP-guarantee can be found, while some level of GMIP privacy can be guaranteed.
>
> > The absence of a discussion on post-processing or the composition rule (in terms of mu) limits the practical applicability and usefulness of the proposed approach.
>
> We would like to highlight that our notion of privacy enjoys powerful post-processing and composition results as discussed in Appendix C.3 of our paper. Due to the hypothesis formulation, these follow from Theorem 4 and Theorem 11 of Dong et al. [1] and are used in Lemma 5.1 of this work. For example, Dong et al. [1, Corollary 2] provide a result which explicitly states that the n-fold composition of $i=1,\ldots, n$ steps which are $\mu_i$-GDP each is $\sqrt{\mu_1^2+ \ldots + \mu_n^2}$-GDP. This result also holds for $\mu$-GMIP.
>
> >  it would be logical to concentrate on scenarios with low FPR. However, the trade-off function is compared across the entire range [0, 1] rather than solely focusing on these low FPR cases
>
> Our theory covers the entire trade-off curve, including arbitrary low FPRs. Following the suggestion of the reviewer, we investigate this regime further and provide the observed trade-off curves in Figures 1 and 2 of the uploaded rebuttal PDF. We see that our bounds hold very well in the low FPR regime. At very low FPR values of $10^{-4}$ the standard errors (across multiple runs) become very large due to the small number of samples, but the empirical estimates always lie well within one standard error.
>
> > The iid assumption is not realistic in practical scenarios, which suggests that the actual privacy guarantees would be lower than those stated in the paper.
>
> Note that the assumption that the distribution of the gradients in subsequent gradient descent steps are independent makes our bounds more conservative (we are thus more cautious which is as desired from a privacy perspective and apply more noise than required and obtain more privacy). This is so as the attacker gets some redundant information about the samples in subsequent steps,  because the sample gradients are correlated as the model and the sample gradients stay very similar through small model updates. This is confirmed in Figure 3b, 3c, where the lower curves indicate that the attack is harder to execute for the attacker than predicted by our theory.
>
> > Since the f-MIP method is specifically designed for SGD, it raises questions about its generalizability to other problem domains.
>
> DP SGD is the workhorse to implement Differential privacy in machine learning [2] and the standard object of study in recent works in privacy-preserving ML literature (e.g., [1,3]). The derivation of potential other mechanisms to implement GMIP is a fruitful avenue for future work.
>
> ------
>
> Thank you for your thoughtful comments and constructive feedback, which we will gladly incorporate into our manuscript. We have addressed the key concerns including the relation between f-DP and f-MIP, composition theorems, and the low-FPR regimes. In light of our response, we kindly ask the reviewer to reconsider the overall assessment of this work.
>
> ------
>
> [1] Dong, J., Roth, A., and Su, W. J. Gaussian differential privacy. Journal of the Royal Statistical Society Series B: Statistical Methodology, 84(1):3–37, 2022.
>
> [2] Abadi, M., Chu, A., Goodfellow, I., McMahan, H. B., Mironov, I., Talwar, K., and Zhang, L. Deep learning with differential privacy. In Proceedings of the 2016 ACM SIGSAC conference on computer and communications security, pp. 308–318, 2016.
>
> [3] Nasr, Milad, et al. "Tight Auditing of Differentially Private Machine Learning." arXiv preprint arXiv:2302.07956 (2023).

---

> > ### Comment · Reviewer_vdUQ · 2023-08-13
> >
> > I appreciate the responses provided by the authors; these have certainly improved my understanding of the paper. Nevertheless, concerning the composition rule, I am still uncertain about how to obtain the same result as proven in GDP. Could you kindly offer a step-by-step heuristic illustration?

---

> > > ### Author Response · Authors · 2023-08-14
> > > **Reply to Reviewer vdUQ**
> > >
> > > Thank you for your kind reply.
> > >
> > > Our general composition result (Lemma 4.1) follows as an application of Corollary 4 from Dong et al., which directly follows from Theorem 11 from Dong et al. Their Theorem does not only hold for f-DP in particular, but more generally as it only relies on the premise that *‘[...] $f$ is a symmetric trade-off function such that $f(0) =  1$ [...]’*. Note that our Gaussian trade-off function from Corollary 5.1 fulfills this condition as it fulfills all criteria of a trade-off function and is symmetric. Hence, we can directly apply Theorem 11 from Dong et al. to our setting.
> > >
> > > We are also happy to provide the outline of the proof of the specific composition result for Gaussian trade-offs which also highlights why their theorems are generally applicable to our trade-off functions. To this end, we consider Theorem 4 by Dong et al., which covers finite compositions (whereas Theorem 11 covers an infinite number of compositions). While this Theorem is stated in terms of $f_i$-DP, note that the $f_i$s are general (not DP specific) trade-off functions that originate from comparing two distributions against one another using hypothesis testing.
> > >
> > > The proof then proceeds in the following steps:
> > >
> > > Consider 2 sequential tests with trade-off functions $f$ and $g$. We want to compose the results from these two tests in the best possible way (i.e., what is the best trade-off the attacker can obtain that features the results from both tests $f$ and $g$). Trade-off functions can equivalently be represented by pairs of distributions that are being tested against each other, e.g., $f = \text{Test}(P, Q)$, where $P$ and $Q$ are distributions.
> > >
> > > Theorem 4 of Dong et al. states that composing tests $f=Test(P,Q)$ and $g=Test(P’,Q’)$ results in the combined trade-off $f \otimes g = \text{Test}(P \times P’, Q \times Q’)$, where $\times$ denotes the joint distribution and $\otimes$ the composed trade-off. As shown in their Lemma C.2, the choice of representation (in terms of $P, Q$) does not matter and has no effect on the result. This part of their proof implies that we can apply their result to testing distributions stemming from both f-DP and f-MIP, as long as we can characterize the tests’ trade-off curves.
> > >
> > > As an example, suppose that we are specifically composing the Gaussian trade-offs $g_{\mu_1}$ and $g_{\mu_2}$ with parameters $\mu_1$ and $\mu_2$. Recall that the $g_{\mu} = \text{Test}(N (0, 1), N (\mu, 1))$ is defined via testing two unit Gaussians of variance one at distance $\mu$ to each other. We can then do the following calculation:
> > >
> > > $g_{\mu_1} \otimes g_{\mu_2} = \text{Test}(N (0, 1), N (\mu_1, 1)) \otimes \text{Test}(N (0, 1), N (\mu_2, 1))$
> > >
> > > $= \text{Test}(N(0, 1)\times N (0, 1),  N (\mu_1, 1) \times N (\mu_2, 1))$
> > > $= \text{Test}(N ((0,0)^\top, \mathbb{I}), N ((\mu_1,\mu_2)^\top, \mathbb{I})$
> > > $= \text{Test}(N ((0,0)^\top, \mathbb{I}), N ((\sqrt{\mu_1^2+\mu_2^2}, 0)^\top, \mathbb{I}) = \text{Test}(N (0, 1) \times N (0, 1), N (\sqrt{\mu_1^2+\mu_2^2}, 1)\times N (0, 1))$
> > > $= \text{Test}(N (0, 1), N (\sqrt{\mu_1^2+\mu_2^2}, 1)) \otimes  \text{Test}(N (0, 1), N (0, 1)) =
> > >  g_{\sqrt{\mu_1^2+\mu_2^2}} \otimes \text{Id} =g_{\sqrt{\mu_1^2+\mu_2^2}}$
> > >
> > > We use the fact that a rotation does not change the hardness of the test. The last line follows from the fact that composition with the $\text{Id}$ test (testing two identical distributions) contains no information (formally proven in Dong et al.) and $\mathbb{I}$ denotes a 2x2 unit matrix.
> > >
> > > We hope that our clarifications regarding the applicability of Theorem 11/Corollary 4 from Dong et al. and the simple outline of the underlying proof for Gaussian compositions clear the reviewer’s uncertainty. Please let us know if you have further questions.

---

> > > > ### Comment · Reviewer_vdUQ · 2023-08-17
> > > >
> > > > I would like to thank the authors for the answer. I have adjusted my overall score based on the responses.

---

### Official Review · Reviewer_Uu8E · 2023-07-06

**Soundness:** 2 fair
**Presentation:** 3 good
**Contribution:** 3 good
**Rating:** 5
**Confidence:** 4

**Summary:**

The authors introduce a new privacy notion called f-membership inference privacy (f-MIP), which relaxes strict Differential Privacy (DP) assumptions thereby promising better model utility. The paper proposes a theoretical analysis of membership inference attacks on DP-SGD based on trade-off curves (similar to f-DP) and introduces a family of f-MIP guarantees called µ-Gaussian Membership Inference Privacy (µ-GMIP)(similar to GDP).  The analysis follows a similar approach to the original DP-SGD analysis: first the privacy budget is derived for single step with subsampling which is then  composed over the training run.

The paper then verifies the theoretical analysis by introducing gradient attacks based on likelihood ratio tests.
The attack requires to know the underlying gradient distribution parameters. They present results for a single DP-SGD step with known parameters and the privacy guarantees seem tight. They also present results for estimated distribution parameters where the guarantees look loser.

**Strengths:**

- The derivations seem sound and follow the well regarded hypothesis testing interpretation in DP
- Investigating the tightness and potential relaxation of DP-SGD is an important problems and the authors make a solid contribution.
- The paper is clearly structured and easy to follow.

**Weaknesses:**

- It would be nice to see an investigation on the validity of the initial assumptions. The paper was motivated that DP is overly conservative since it also holds for pathological datasets such as empty datasets and singletons with an adversarial sample. While this is true, it has been recently shown that a simple canary insertion in an otherwise natural dataset may be sufficient to produce tight lower bounds for DP accountants [Nasr et al 2023].
- It would be great to extend the FPR and TPR ranges in the plots in figure 4 to smaller values. Ideally all the way to the first data point. These ranges capture very relevant adversary objectives where an adversary only cares about identifying one sample but that with high confidence.
- Minor:
  - Only asymptotic guarantees for composition. Recently, there has been significant progress in numerical composition of DP guarantees.


**References**

Nasr, Milad, et al. "Tight Auditing of Differentially Private Machine Learning." arXiv preprint arXiv:2302.07956 (2023).


**Questions:**

- Is it assumed that $\tau$ in the noise parameter includes the batch size? Algorithm 1 does not scale the noise by the batch size which is different to typical DP-SGD [Abadi et al 16].
- Figure 4a. Purchase and CIFAR10 seem to exceed the analytical bound for low FPRs. Is this because of the earlier mentioned assumption that the challenge points are sampled from the distribution which for low FPRs may be already distributional outliers?
- Are the techniques for composing DP guarantees numerically also applicable in this setting?

**Limitations:**

As discussed in weaknesses. I believe the discussion about the validity of the initial assumptions is limited.

---

> ### Author Rebuttal · Authors · 2023-08-07
>
> We thank the Reviewer for their thoughtful review and were pleased to hear that the Reviewer found our paper to be *clearly structured* and to be a *solid contribution*. We will answer the individual points raised below.
>
> > It would be nice to see an investigation on the validity of the initial assumptions. [...] it has been recently shown that a simple canary insertion in an otherwise natural dataset may be sufficient to produce tight lower bounds for DP accountants [Nasr et al 2023].
>
> The reviewer’s observation is accurate. The problem of finding a tight lower bound for DP is considered solved as canary insertions (i.e., outlier sample insertions to datasets) make the DP bound tight (Nasr et al 2023). Therefore, to improve the utility over DP-trained models, new takes on privacy are in demand where one maps the exact threat model to a suitable privacy notion, which is precisely what our work accomplishes. If we can rule out pathological canary insertions (e.g., because the adversary, by construction, has no dataset access), other notions than standard differential privacy can be considered. This is what has recently been identified as a fundamental open problem by a large group of leading researchers from the field of privacy preserving ML ([1, sections 2.1, 4.1, 4.3].
>
> We explicitly compare the assumptions underlying DP and MIP in new table in the general comment. Regarding the validity of our theoretical assumptions, we conduct 4 ablation studies in the rebuttal PDF file (Figure 3). In all these cases, we find that our theoretical results accurately predict attack success.
>
> > It would be great to extend the FPR and TPR ranges in the plots in figure 4 to smaller values
>
> We thank the reviewer for this constructive suggestion. We provide the curves in Figures 1 and 2 in the uploaded rebuttal PDF (Figure 1 shows individual runs up to the first data point). Our bounds hold very well in the low FPR regime. At very low values for the FPR, the standard errors (across multiple runs) become very large due to the small number of samples, but the empirical estimates always lie well within one standard error.
>
> > Is it assumed that $\tau$ in the noise parameter includes the batch size? Algorithm 1 does not scale the noise by the batch size which is different to typical DP-SGD [Abadi et al 16].
>
> This is correct. Different works use different parameterization of the noise (e.g., Abadi et al. and Dong et al. [3] also use different parameterizations). As in our paper, we denote the dimension-wise variance of the total noise that is added to the average gradient by $\tau^2$. Thus, our parameter $\tau$  can be converted to the $\sigma$ used by Abadi et al. as follows: $\tau^2 =  \frac{\sigma^2_{\text{Abadi}}C^2}{n}$. If it would make the manuscript more accessible to the reviewer, we will be happy to express the added noise through the parameterization used by Abadi et al. in our manuscript.
>
> > Figure 4a. Purchase and CIFAR10 seem to exceed the analytical bound for low FPRs. Is this because of the earlier mentioned assumption that the challenge points are sampled from the distribution which for low FPRs may be already distributional outliers?
>
> Our results cover the entire trade-off curve including the low-FPR regimes. Our theoretical predictions are probabilistic and our empirical results are always within the statistical standard error. To make this more clear, we provide plots using more samples in the rebuttal PDF file. On the setup corresponding to Figure 4a in the main paper (Figure 2a in the rebuttal PDF) our results always reside within the standard error and match the theoretical prediction almost exactly at the earlier limit of $10^{-2}$. At very low values of FPR of $10^{-4}$ the standard errors (across multiple runs) become very large due to the small number of available samples to estimate the FPRs and TPRs.
>
> > Are the techniques for composing DP guarantees numerically also applicable in this setting?
>
> We would like to highlight that our privacy notion already comes with powerful composition and post-processing results that were transferred from the work of Dong et al. [3]. We agree that the asymptotic result under subsampling and composition presented in Lemma 4.1. may potentially be improved using numeric estimation techniques similar to those proposed by Gopi et al. [2], which is a great suggestion for follow-up work.
>
> >  I believe the discussion about the validity of the initial assumptions is limited.
>
> To further address this point, we provide a table detailing the different assumptions underlying various attacks and privacy notions in the general comment, which we will gladly include in our final manuscript.
>
> -------------------
>
> We were happy to hear that the Reviewer found our work to be solid and sound in general, as mentioned in your review. We sincerely hope that we have addressed your remaining concerns regarding the low-FPR regime and the discussion of the initial threat models, leaving no major concerns unaddressed. In light of this response and your positive assessment, we would kindly ask you to consider updating the review score in light of this rebuttal and the changes listed in the general response.
>
> ----------------
>
> **References**
>
> [1] Challenges towards the Next Frontier in Privacy, Cummings et al (2023),  arXiv:2304:06929
>
> [2] Sivakanth Gopi, Yin Tat Lee, and Lukas Wutschitz: Numerical Composition of Differential Privacy, arXiv:2106.02848
>
> [3] Dong, J., Roth, A., and Su, W. J. Gaussian differential privacy. Journal of the Royal Statistical Society Series B: Statistical Methodology, 84(1):3–37, 2022.

---

> > ### Comment · Reviewer_Uu8E · 2023-08-12
> >
> > Thank you for the detailed clarification and including the low FPR plots.
> >
> > I think the table is a great addition, however, I think that some statements about the usefulness and the relevance of the threat model are overstated.
> >
> > > The sample $x'$ for which membership is to be inferred is drawn from the data distribution $D$. Therefore, MI is concerned with typical samples that can occur in practice
> >
> > I would argue that inserting a worst case sample is an easy task for an adversary in practice e.g. Census data, recommendation datasets, language modelling datasets. However, I admit that there are applications where the threat model in this paper holds and worst case samples are hard to insert e.g. medical datasets, etc.
> >
> > I think a clear discussion of when the proposed threat model is applicable and **also when not** would strengthen the paper. The table is a great first step.

---

> > > ### Author Response · Authors · 2023-08-14
> > > **Reply to Official Comment by Reviewer Uu8E**
> > >
> > >
> > > Thank you for your positive reply. We agree with the reviewer’s observations that the threat model is useful for some applications, but may not always be applicable as it requires an adversary with restricted dataset access:
> > >
> > > - In financial and healthcare applications, the data is often collected from actual events (e.g., past trades) or only a handful of people (i.e., trusted hospital staff) have access to the records. In such scenarios, it might be overly restrictive to protect against worst-case canary attacks as attackers cannot freely inject arbitrary records into the training datasets.
> > > - In other cases such as online surveys and census data, as mentioned by the Reviewer, an attacker may indeed be able to do sample injection. These attacks are not covered by $f$-Membership Inference Privacy and a more general notion such as Differential Privacy should be preferred.
> > >
> > >
> > > We will happily add these clarifications regarding to our paper. We will also extend the table by another line with recommended example usages:
> > >
> > > |                           | $f$-Differential Privacy                                                  | $f$-Membership Inference Privacy                                                                                                                                                                 |   |   |   |   |   |   |   |
> > > |---------------------------|---------------------------------------------------------------------------|--------------------------------------------------------------------------------------------------------------------------------------------------------------------------------------------------|---|---|---|---|---|---|---|
> > > | Best used in applications | where specific attack model is unknown. Offers a form of general protection. | where dataset access (e.g. canary injection) of an attacker can be ruled out and the main attack goal lies in revealing private training data (e.g., membership inference, data reconstruction). |   |   |   |   |   |   |   |
> > > |                           |                                                                           |
> > >
> > > Let us know what you think. We would be happy to incorporate further suggestions!

---

> > > > ### Comment · Reviewer_Uu8E · 2023-08-14
> > > >
> > > > Thanks. I think this addition will be help the reader navigate the space. I've increased my overall score.

---

### Official Review · Reviewer_PQQ4 · 2023-07-09

**Soundness:** 3 good
**Presentation:** 3 good
**Contribution:** 3 good
**Rating:** 4
**Confidence:** 3

**Summary:**

the submission extends the work on privacy guarantees specifically for protecting data membership inference attacks, and the extension is on using the Gaussian distribution to characterize the trade-off function. The proposed Gaussian Membership Inference Privacy has the same parametrization as the Gaussian Differential Privacy, but provides weaker privacy guarantees in the sense that an attacker only has access to the learned machine learning model and true data distribution or the public data.

The submission further shows that a single step of noisy SGD is approximately Gaussian membership Inference privacy with a privacy parameter $\mu$, and empirically shows that the theoretical trade-off curve is the upper bound of the ROC curves of practical attacks with the given public knowledge.

**Strengths:**

1. the membership inference attack captures a class of realistic attacks against machine learning models, where the attacker can't manipulate the private training data but has access to the data distribution and the learned model through an API. It indeed provides weaker privacy guarantees than Differential Privacy does, but studying it might lead us to a more practical privacy definition for protecting individuals in machine learning systems.

2.  one-step noisy SGD can be captured by f-membership inference privacy, and can be captured approximately by Gaussian Membership Inference privacy.

3. a very important observation is that averaging over norm-clipped gradients satisfies GMIP with a bounded privacy parameter, which means that it already protects, to some extent, against membership inference attacks on gradients.

**Weaknesses:**

1. Gaussian Differential Privacy exactly captures the trade-off function of the Gaussian mechanism, which is used to release noisy gradients. However, Gaussian Membership Inference Privacy only approximately captures so, and I am wondering what 'approximately' entails, and whether it is actually useful compared to f-membership inference privacy.

2. the assumption for the algorithm that an attacker would use is that the algorithm has to return binary answers, but we've seen successful attacks using ranking, for example, the watchdog experiments in [1]. Thus, I am questioning this assumption on top of an attacker's algorithm.

3. I was hoping to see experiments with real machine learning models with only API access to them, e.g. an attacker can only have access to the trained classification neural network through the final probabilistic predictions over classes. The distributional assumption over the susceptivity does not hold anymore, and it is interesting to see how the proposed privacy definition captures the success of membership inference attacks.



[1] https://www.pnas.org/doi/epdf/10.1073/pnas.2218605120

**Questions:**

please see the weaknesses.

**Limitations:**

please see the weaknesses.

---

> ### Author Rebuttal · Authors · 2023-08-07
>
> We are very grateful for the positive comments, which highlight the relevance of the studied Membership Inference threat model, our ability to analytically capture SGD steps with f-MIP, and which highlight the finding that we can obtain MIP through averaging of gradients only. We will address the remaining points below.
>
> > Gaussian Membership Inference Privacy only approximately captures so, and I am wondering what 'approximately' entails, and whether it is actually useful compared to f-membership inference privacy.
>
> First note that our exact formulation from Theorem 5.1 contains the CDF of a $\chi^2$ distribution, while our approximation from Corollary 5.1 uses the Gaussian CDF. In general, it is well known that for large $d$, the $\chi^2$ distribution converges to a Gaussian distribution at a convergence rate of $\mathcal{O}(1/\sqrt{d})$ (see Theorem 1 in [7], see Figure 2a of the paper for an illustration). Regarding $n$, we can apply the Berry-Esseen Theorem to get a finite sample bound for the error between the true averaged gradient distribution and the Gaussian distribution. The finite sample error depends on a universal constant (some number depending on third moments), and scales with $\mathcal{O}(1/\sqrt{n})$. Hence, as we increase the number of gradient vectors in a batch, we expect the error to decrease. To demonstrate that this is in fact the case, we have added ablation results in Figure 3 of the rebuttal PDF file for small dimensions (d) and for small batch sizes (n). In these experiments, we start with small values of $d$ and $n$ and increase these which empirically demonstrates that our approximations are accurate even for small $d$ and $n$ despite our theory being carried out for larger sample sizes.
>
> > the assumption for the algorithm that an attacker would use is that the algorithm has to return binary answers, but we've seen successful attacks using ranking, for example, the watchdog experiments in [1]. Thus, I am questioning this assumption on top of an attacker's algorithm.
>
> Membership Inference attacks are a common standard in the literature (e.g., [1-6]) and are one of the simplest attacks that can be carried out by an adversary. The assumption about the output of the membership inference attack follows the standard definition originally introduced by [4], which has since been adopted in many popular prior works (e.g., [1,5]). The reference provided by the reviewer considers reconstruction attacks, which are different in the sense that full samples are reconstructed. As opposed to reconstruction attacks, membership inference attacks attempt to only verify whether a given sample was part of the training dataset. Due to the simplicity of the MI attack, protecting against MI should also offer protection against other, more complex attacks as well. For example, one could cast the data reconstruction attack as a series of sequentially applied membership inference attacks where the task consists of verifying whether a given token was part of the training data set. We will gladly include a discussion of reconstruction attacks such as the one mentioned in our manuscript.
>
> > I was hoping to see experiments with real machine learning models with only API access [...] it is interesting to see how the proposed privacy definition captures the success of membership inference attacks.
>
> The membership inference threat model in our work uses access to gradients, which is a common scenario in federated learning setups. Due to the use of the gradients our attack is stronger than attacks that rely solely on API access. To showcase the superiority of our attack, we run the state-of-the-art API-only attack suggested by Carlini et al. [1] on real machine learning models and obtain the following results shown in Figure 5 of the Appendix. These results show that our attack (Figure 4) is 2-10 times stronger than the state-of-the-art attack based on API access only. Thus, adding the noise required to protect under our optimal attack under gradient access, also protects against attacks that rely on API access only.
>
> -------
>
> We appreciate the positive comments by the reviewer and are glad to hear that the Reviewer appreciated the “Soundness”, “Presentation”, and “Contribution” of our work by awarding a rating of "Good". We hope that we have addressed the remaining concerns with the new ablation studies and the results on the state-of-the-art API-only attack by Carlini et al. [1]. Given the positive evaluations and our clarifications, we would kindly ask the Reviewer to reconsider their overall score.
>
> -----
>
> **References**
>
> [1] Nicholas Carlini, Steve Chien, Milad Nasr, Shuang Song, Andreas Terzis, and Florian Tramèr. Membership inference attacks from first principles. IEEE Symposium on Security and Privacy (SP), 2022.
>
> [2] Jiayuan Ye, Aadyaa Maddi, Sasi Kumar Murakonda, and Reza Shokri. Enhanced membership inference attacks against machine learning models. ACM CCS, 2022.
>
> [3] Reza Shokri, Marco Stronati, Congzheng Song, and Vitaly Shmatikov. Membership inference attacks against machine learning models. In 2017 IEEE symposium on security and privacy (SP), IEEE, 2017.
>
> [4] Samuel Yeom, Irene Giacomelli, Matt Fredrikson, and Somesh Jha. Privacy risk in machine learning: Analyzing the connection to overfitting. In 31st IEEE Computer Security Foundations Symposium, 2018..
>
> [5] Martin Pawelczyk, Himabindu Lakkaraju, Seth Neel. On the Privacy Risks of Algorithmic Recourse. Proceedings of The 26th International Conference on Artificial Intelligence and Statistics (AISTATS), 2023.
>
> [6] Christopher A. Choquette-Choo, Florian Tramèr, Nicholas Carlini, and Nicolas Papernot. Label-only membership inference attacks. In Proceedings of the 37th International Conference on Machine Learning (ICML), 2020.
>
> [7] Donagh Horgan and Colin C. Murphy. On the Convergence of the Chi Square and Noncentral Chi Square Distributions to the Normal Distribution. IEEE Communications Letters, Vol. 17, No. 12, 2013

---

### Official Review · Reviewer_qYbt · 2023-07-10

**Soundness:** 3 good
**Presentation:** 3 good
**Contribution:** 2 fair
**Rating:** 5
**Confidence:** 3

**Summary:**

Authors came up with a privacy definition that is more relaxed than DP. It is called Gaussian Membership Inference Privacy (GMIP) which consists of a hypothesis testing which is supposed to decide whether a single instance is present in the training data. DP implies GMIP. The proposed privacy framework is applied with SGD and in experiments found to be much better than DP in terms of utility.

**Strengths:**

* There is a need to come up with some privacy definition that is more practical than DP.
* Research direction is promising.

**Weaknesses:**

* I believe that standard notions from statistical hypothesis testing are reinvented, and the results seems not so surprising taken into account results already published in testing. I would suggest to reuse those results (if it is possible). Nevertheless, please address my questions!

**Questions:**

* Definition 4.1 is basically the most powerfull test. $\mathcal{E}$ is the significance and $\beta$ is the power function, or more concretely 1 - power function?
* Regarding Theorem 4.1: according to the Neyman-Pearson fundamental lemma the risk function is always convex when the null and alternative consist of a single distribution. And the testing problem presented in (3) is like that. See Lehamnn-Romano: Testing Statistical Hypotheses, Sec 3.2.
* Testing are applied in many rounds and i guess only union bound is used over the individual SGD steps. Some sequential hypthesis testing might be worth to consider?


**Limitations:**

I believe that this research direction is interesting and promising. However, convexity of risk function seems not novel observation which is the key of having most uniformly powerful test. And this is what the paper relies on. I do not understand why to stick to Gaussion distribution, since any distribution from the exponential family can be used in a very similar way. Furthermore, FNR are controlled in each SGD step independently from each other however sequential testing approaches might applied here which would make this paper much more interesting. So technical contribution is somewhat limited.

---

> ### Author Rebuttal · Authors · 2023-08-07
>
> We thank the Reviewer for their thoughtful review and the questions raised. We will clarify the individual points below.
>
> > I believe that standard notions from statistical hypothesis testing are reinvented, and the results seems not so surprising taken into account results already published in testing. [...]
>
> We would like to highlight that we only *leverage* standard notions from statistical hypothesis testing. However, we frame the problem of membership inference privacy as a hypothesis testing problem to be able to use testing tools to develop a constructive theory. This theory helps us set up a privacy notion geared towards the realistic and empirically well studied threat model of membership inference attacks and a constructive algorithm which effectively protects against such attacks. Such results are highly sought after in the privacy literature, where the need to reconsider threat models has recently been proclaimed by leading researchers in the field [1, sections 2.1, 4.1, 4.3].
>
> > Definition 4.1 is basically the most powerfull test. $\mathcal{E}$ is the significance and $\beta$ is the power function, or more concretely 1 - power function?
>
> The reviewer is right in that Definition. 4.1 is reminiscent of the most powerful test. However, there are several differences to the standard definition of the most powerful test that are important in our work and that motivate the need for Definition 4.1. Most prominently, we adjust the definition of the most powerful test (i.e., hypothesis tests) to be applicable to the membership inference problem (this is what definition 4.1 accomplishes). Please note that a straightforward construction of the most powerful test does not work in this setup. This is the case because the adversary does not only run one hypothesis test to figure out whether one sample belongs to the training data set or not; instead, the adversary draws samples $x$ and runs individual, sample-dependent and different hypotheses tests for each drawn sample. This is necessary due to the formulation of the distribution $A_1(x)$ under the alternative hypotheses in the formulation of the test (Eqn. 3), which depends on the sample $x$. The value of $x$ is known to the adversary. We therefore require a tool to compose the results from the different hypothesis tests, which we carefully craft in Definition 4.1.We can then compute the expected trade-off curve for an adversary that runs tests with different powers according to the observed samples $x$.
>
> > Regarding Theorem 4.1: according to the Neyman-Pearson fundamental lemma the risk function is always convex when the null and alternative consist of a single distribution. And the testing problem presented in (3) is like that. [...]
>
> In Definition 4.1., the distributions are not simple, but instead depend on a stochastic sample $x$. In particular, the distribution under the alternative hypotheses, $A_1(x)$ in Eqn. (3), depends on the individual sample $x$, which is known to the attacker and can be used to run sample-specific tests. Therefore, to compute the expected trade-off curve that can be reached by an attacker who samples $x$, we use the trade-off defined in Definition 4.1. While its properties shown in Theorem 4.1 may seem intuitive, the proof is not trivial, which is why we decided to include it for completeness. Note that we consider our main contributions to be Theorem 5.1 and Corollary 5.1, which are the first of its kind and precisely quantify the factors that lead to successful Membership Inference.
>
> > Testing are applied in many rounds and i guess only union bound is used over the individual SGD steps. Some sequential hypthesis testing might be worth to consider?
>
> While we do not use the union bound for the composition as suggested by the reviewer, our result composing multiple steps of SGD given in Lemma 5.1 and follows from Theorem 11 and Theorem 4  in Dong et al. [2]. These results already provide tight composition bounds for hypothesis tests that satisfy all our needs. We are however confident that it would be possible to derive similar results using union bounds as well.
>
> > I do not understand why to stick to Gaussion distribution, since any distribution from the exponential family can be used in a very similar way
>
> Please note that we don’t *assume* the Gaussian distribution. Instead, we consider averages over parameter gradients commonly used in minibatch stochastic gradient descent, which then follow a Gaussian distribution by the central-limit theorem (CLT). We put no distributional assumption's on the gradients at all.
>
> > Furthermore, FNR are controlled in each SGD step independently from each other however sequential testing approaches might applied here which would make this paper much more interesting. So technical contribution is somewhat limited.
>
> Note that we use tight results from the testing literature, which consider the full trade-off curve of a sequence of hypothesis tests (see, e.g., Theorem 4 of Dong et al. [2]). These bounds are optimal for any overall FNR and also cover cases where, e.g., different FNRs at targeted at the individual tests or the statistics of different tests are combined in non-trivial ways.
>
> Overall, we would like to stress that we *use* tools from the hypotheses testing literature, but this is not our key contribution. The key contribution of our work lies in defining MI privacy through the composed trade-off curve and bounding the attack risk of DP-SGD with respect to membership inference attacks. We will adjust our manuscript to better differentiate our contributions from the existing tools that we use. We hope that our replies have clarified the matter and are happy to answer any follow-up questions.
>
> **References**
>
> [1] Challenges towards the Next Frontier in Privacy, Cummings et al. (2023), arXiv:2304:06929
>
> [2] Dong, J., Roth, A., and Su, W. J. Gaussian differential privacy. Journal of the Royal Statistical Society Series B: Statistical Methodology, 84(1):3–37, 2022.

---

> > ### Comment · Reviewer_qYbt · 2023-08-17
> > **Thanks for the rebuttal.**
> >
> > Authors addressed my concerns properly. I liked the topic of the paper because to better understand inference attacks is a very timely question. Therefore I recommend the paper be accepted.

---

### Official Review · Reviewer_DVNo · 2023-07-23

**Soundness:** 2 fair
**Presentation:** 3 good
**Contribution:** 3 good
**Rating:** 5
**Confidence:** 4

**Summary:**

The paper proposes a notion of Gaussian membership inference privacy (GMIP) to capture the information leakage of a training algorithm about a data point $x$ when all the remaining training datasets are randomly drawn from a data distribution. The new GMIP definition has two main benefits compared to the prior leakage definition.
1. It captures the entire trade-off curve of membership inference as a hypothesis test and is thus more informative than prior average-case privacy definitions such as membership inference advantage.
2. It stochastically composes the trade-off curves over the randomly drawn remaining training dataset (other than the target data). It, therefore, captures leakage against a more realistic adversary that does not have control over the remaining training dataset (compared to the worst-case f-DP definition).

Following this new GMIP definition, the paper analytically proves GMIP for the DP-SGD algorithm via an analytical likelihood ratio attack on an observed noisy gradient (under assumptions on the gradient distribution, model dimension, and dataset size). The proved GMIP bound has interesting dependencies on the model dimension and a data-dependent constant associated with the gradient distribution. To further illustrate the tightness of this GMIP bound, the authors evaluate the performance of various attacks for a mu-GMIP noisy SGD algorithm and show when the attack performances are close to the GMIP upper bound. Finally, the authors evaluate model accuracy under mu-GMIP and show that it improves over the model accuracy under mu-GDP, which indicates a privacy-utility trade-off gain due to relaxed adversary assumption.

**Strengths:**

- The paper studies an important problem of analytically bounding informative information leakage of training algorithms against realistic adversaries.
- The critical component is a novel analytical derivation of the likelihood ratio test, assuming that the aggregated noisy gradients follow a multivariate Gaussian distribution.
- The proved GMIP bound is novel and has interesting dependencies on various factors about the model and data distribution. The discussion about the tightness of the bound is detailed and supported by empirical evaluations.

**Weaknesses:**

[W1] The main weakness is that the proved GMIP bound is based on several approximation arguments and relies on assumptions about the gradient distribution, the model dimension, and the dataset size. The authors should clarify these approximations and assumptions in the comparison in Figure 1. Otherwise, the comparison to the mu-GDP algorithm is unfair or misleading.

[W2] Specifically, one of the assumptions used in approximating the LRT (line 618 appendix D.1) is that the distribution of averaged gradient follows a Gaussian distribution, provided that the number of averaged samples is large enough. This assumption is invalid because the gradients are clipped before averaging, so the distribution of averaged gradient is bounded and is not close to a Gaussian distribution.

[W3] The proved GMIP bound Theorem 4.2. grows indefinitely with regard to model dimension d. Such dimension dependency does not exist in the standard mu-GDP bound for DP-SGD (which implies GMIP). This suggests that the GMIP bound may be less tight than the standard mu-GDP bound for the high-dimensional problem, which is the case for training a deep neural network.

**Questions:**

[Q1] What exactly are the approximations and assumptions used for the proved GMIP bound, for the comparison figure 1? How much would possible issues with assumptions (such as the one mentioned in weakness [W2] and insufficiently large dataset size n and model dimension d) break the proved GMIP bound?

[Q2] Is the proved GMIP bound less tight than the standard mu-GDP bound for DP-SGD, when the model dimension is large? See weakness [W3] for more information.

**Limitations:**

The limitations regarding assumptions made for the analysis should be discussed more. See weakness [W1] and [W2] for more details.

---

> ### Author Rebuttal · Authors · 2023-08-07
>
> We thank the Reviewer for their thoughtful review and the questions raised. We clarify the individual points below.
>
> > [W1] The main weakness is that the proved GMIP bound is based on several approximation arguments and relies on assumptions about the gradient distribution, the model dimension, and the dataset size.The authors should clarify these approximations and assumptions [...]
>
> We clarify that we do not make parametric approximation assumptions about the gradient distribution. Our theoretical results stem from the application of the central limit theorem (CLT) to averages formed over mini-batches of gradients and do *not* use parametric assumptions on the gradient distribution. Loosely speaking, the CLT states that the distribution of sample means converges to a Gaussian for large enough sample sizes regardless of the distribution of individual gradients. Therefore, we dont need to make assumptions about the shape of the gradient distribution. For a discussion on other parameters see the response to [Q1].
>
> We provide a full discussion of threat models of other privacy notions in the general comment due to space constraints, and will incorporate them into the final version of our manuscript.
>
> > [W2] Specifically, one of the assumptions used in approximating the LRT (line 618 appendix D.1) is that the distribution of averaged gradient follows a Gaussian distribution [...] This assumption is invalid because the gradients are clipped before averaging, so the distribution of averaged gradients is bounded and is not close to a Gaussian distribution.
>
> While it is correct that the gradients are clipped, we stress that, in non-technical terms, the Central Limit Theorem states that the **distribution of sample means converges to a normal distribution for large enough sample sizes, regardless of the shape of the distribution of the individual samples** [1, p.66-68]. This means that the individual gradients may be clipped and that the CLT can still be applied to averages over clipped gradients. We demonstrate this empirically through an ablation study (see answer to [Q1])
>
> > [W3] The proved GMIP bound Theorem 4.2. grows indefinitely with regard to model dimension d. [..] This suggests that the GMIP bound may be less tight than the standard mu-GDP bound for the high-dimensional problem, which is the case for training a deep neural network.
>
> > [Q2] Is the proved GMIP bound less tight than the standard mu-GDP bound for DP-SGD, when the model dimension is large? [...]
>
> Indeed there might exist settings where our results require more noise for $\mu$-MIP than $\mu$-DP. In these cases one should resort to $mu$-DP as it implies $mu$-MIP. This is however not the case for the realistic models considered in this work.
>
> The dependency on the parameter $d$  is a consequence of our intentionally broad proving strategy. Our proof approach consists of two key steps: First, we establish an optimal LRT framework under general gradient distributions, without imposing any cropping constraints (App. D1). This initial step serves as the foundation for our subsequent analysis and is (1) as general as possible, i.e., no distribution assumptions and (2) optimal in the sense of the Neyman-Pearson Lemma, i.e., it cannot be improved. Our result covers all models trained with standard SGD and is remarkable in its generality as it is the first to suggest clear conditions when adding noise is not required to reach $mu$-MIP.
>
> Second, we specialize our findings to noise addition on cropped random variables (App D2). This analysis may potentially be improved. We offer the following intuitive rationale: Since variance of noise is considered fixed across dimensions, the introduction of additional dimensions naturally leads to an overall increase in norms, approx. on the order of $O(sqrt(d))$. It's important to note that the dependence on $d$ may thus be removed through full incorporation of the fixed cropping threshold $C$ as is common in DP. We leave this improvement for future work as it requires a substantially more complicated analysis
>
> Finally, we emphasize that our work already achieves a significant milestone by being the first to analytically characterize the entire trade-off curve for membership inference attacks.
>
> > [Q1] What exactly are the approximations and assumptions used for the proved GMIP bound, for the comparison figure 1? How much would possible issues with assumptions [...] break the proved GMIP bound?
>
> While we require that the gradient dim. $d$ and batch size $n$ are sufficiently large, we put these aspects into perspective. Following the suggestion of the Reviewer, we ran four ablation studies in the rebuttal PDF to investigate the effects of the params. on the GMIP bound. We do this by averaging gradients that follow a Gaussian or a uniform distribution and observe that the $\mu$-GMIP bound closely reflects the empirically observed trade-off curves for:
>
> *   Realistic values of batch size $n$ ($n >= 10$) both when the Gradients follow a Gaussian (Figure 3a, e) or a Unit distribution (Figure 3b, f)
> *   A wide range of cropping thresholds C from 1 to 10 (in this example, the expected value of $\lVert \theta_i \rVert = 5$, Figure 3c, g). We see no effect on the validity of our results (see CLT argument).
> *   Small values of $d$. Our bound is an extremely good approximation even for values as small as $d=2$ (Figure 3d, h)
>
> ---
>
> We thank the Reviewer again for the feedback, which will certainly help improve our manuscript. We sincerely hope that we have particularly addressed your main concern of why the CLT is in fact applicable and why no distributional assumptions are required, which are empirically confirmed in the new ablation experiment provided. We are happy to include these remarks in the final version of the paper. We would kindly request the Reviewer to reconsider the review score in light of this response.
>
> Reference
>
> [1] Y. Dodge. The Concise Encyclopedia of Statistics. Springer, 2008

---

> > ### Comment · Reviewer_DVNo · 2023-08-13
> > **Thanks for the rebuttal**
> >
> > Thanks the authors for clarifying their CLT argument for the mean of clipped gradient distribution. It definitely clarifies my doubt about the application scope of the bound. However, the CLT argument is an approximation that is only exact at infinite $n$. The fact that the authors are analyzing the converged Gaussian distribution due to CLT argument makes their current upper bound invalid for any finite $n$ with certain probability. To this end, a correct bound would require error correction terms related to using CLT (similar to [Theorem 3.7, Dong et al.]). I'd like to request the authors to update the statement of their GMIP bound to either add explicit error correction terms due to approximations, or add precise descriptions of the assumptions required for the bound to hold (e.g. the gradient mean exactly follows the Gaussian distribution).
> >
> > Another remaining concern (which is direct consequence of the approximation error mentioned above) is that the proved privacy bound grows with model dimension, and is less tight than the dimension-independent Gaussian DP bound for DP-SGD under large model dimensions. (As the authors acknowledge in the rebuttal.)
> >
> > I still find the work interesting, and as the authors point out, it is the first-time that a trade-off function under MIA is analytically estimated. However, it also has the above two important limitations and therefore I'm keeping my score still as borderline accept.

---

> > > ### Author Response · Authors · 2023-08-14
> > > **Reply to Reviewer DVNo**
> > >
> > > Thank you for your thoughtful reply.
> > >
> > > We will be happy to make this point more clear in our manuscript. We will also discuss how Berry-Esseen’s theorem, which yields an error of order $\mathcal{O}(1/\sqrt{n})$, might be used to bound the error. It is also worthwhile to check whether the bound of Dong et al. (pointed out by the reviewer) or its proving technique are applicable. Their bound provides an even faster convergence rate of $\mathcal{O}(1/n)$. Finally, we would like to stress that our empirical studies suggest that the error seems to be negligible even for moderate $n$ in the general and in the relevant low-FPR regime.
> > >
> > > Thank you for your time and your continued support of our work.

---

### Author Rebuttal · Authors · 2023-08-09

We thank all reviewers for their constructive feedback, which allows for direct improvements of our work. Inspired by the reviewer comments, we plan to make the following changes and intend to use the additional page provided in the camera-ready version for the following additions:

   * We add a table comparing the assumptions and premises of  f-DP and our f-MIP privacy notion (shown below). We will additionally compare our suggested class of new membership inference attacks with other attacks from literature.
   * We empirically investigate the low-FPR regime using additional plots up to a FPR of $10^{-4}$ and find our empirical results to match our theory well.
   * We ran 4 new empirical studies on the effect of the size of the parameter vector $d$, the number of samples $n$, the cropping threshold $C$ and the type of gradient distribution. They confirm that 1) our theoretical bounds are highly accurate across parameter ranges, 2) that the CLT can in fact be applied to means of cropped random variables (addressing Reviewer ``DVNo``s concern) and 3) that our theory holds with minor ramifications when $n$ and $d$ are extremely small. To explain this (surprising) behavior we provide theoretical insights on the convergence of the errors  in the response to Reviewer ``qYbt``.
   * In Response to Reviewer ``vdUQ``, we provide a new Corollary relating the values of the parameter $\mu$ between $\mu$-GDP and $\mu$-MIP.

We would additionally like to stress that our work makes *several impactful contributions*:

1. We are the first to formulate membership inference as a hypothesis testing problem and use the trade-off function of the test to define a versatile notion of f-Gaussian Membership Inference Privacy. Such results are highly sought after in the privacy literature, where the need to reconsider threat models has recently been proclaimed by leading researchers in the field [1, sections 2.1, 4.1, 4.3].
2. Our hypothesis test formulation is valuable, as it allows for a fine-grained theoretical analysis of membership inference attacks. Our main contributions are Theorem 5.1 and Corollary 5.1, which precisely quantify the factors that lead to membership inference attack success in a step of stochastic gradient descent. Our results rely on constructing the most powerful test and are remarkably general: They also cover all ML models trained with standard gradient-based optimization, even without noise or gradient cropping.
3. Notably, through this formulation we can transfer composition and post-processing results from existing literature. We apply those to steps of noisy stochastic gradient descent to bound the attack success for training of an entire model
4. Based on our theoretical insights, our final contribution consists of a constructive algorithm that quantifies the required noise level in SGD to defend against membership inference attacks.

Thank you again for your feedback. We will be happy to answer any further questions.



-------------------------


**Table: Comparing $f$-Differential Privacy and $f$-Membership Inference Privacy**

|                     | $f$-Differential Privacy                                                                                                                                                                    | $f$-Membership Inference Privacy                                                                                                                                                 |
|---------------------|---------------------------------------------------------------------------------------------------------------------------------------------------------------------------------------------|----------------------------------------------------------------------------------------------------------------------------------------------------------------------------------|
| Adversary goal      | Distinguish between $D$ and $D^\prime$ for any $D$, $D^\prime$ that differ in at most one instance                                                                                          | Distinguish whether $x^\prime \in S$ (training data set) or not.                                                                                                                 |
| Data access         | Attacker has full dataset access. For example, the attacker can poison or adversarially construct datasets on which ML models could be trained; e.g., $D = \\{ \\}$ and $D^\prime = \\{100000\\}$ | Attacker has no access to the training data set; i.e., the model owner privately trains their model free of adversarially poisoned samples.                                      |
| Protected instances | The instance in which $D$ and $D^\prime$ differ is arbitrary. This includes OOD samples and extreme outliers                                                                                | The sample $x^\prime$ for which membership is to be inferred is drawn from the data distribution $D$. Therefore, MI is concerned with typical samples that can occur in practice |
| Model knowledge     | The attacker knows the full model architecture including hyperparameters and has full access to samples from the distribution and the parameters during training                             | The attacker knows the full model architecture including hyperparameters and has full access to samples from the distribution and the parameters during training                  |


**References**

[1] Challenges towards the Next Frontier in Privacy, Cummings et al (2023),  arXiv:2304:06929

---

> ### Author Response · Authors · 2023-08-21
> **Update**
>
> As the discussion period comes to a close, we would like to thank the reviewers for actively participating in the discourse and, where relevant, for their thoughtful reevaluation of the overall scores. We are fully committed to addressing the points discussed, which allow for valuable improvements to our manuscript. Let us know if you have any further questions.

---

### Decision · Program_Chairs · 2023-09-21

**Decision:**

Accept (poster)

**Comment:**

The reviewers appreciate the paper's insightful approach to analyzing information leakage in training algorithms, particularly praising the new GMIP bound as a significant addition to the field. Although there seems to be some overlap with existing concepts from statistical hypothesis testing, the fresh perspectives and thorough empirical evaluations give the paper a strong position. The meta-reviewer recommends acceptance as a poster, anticipating that a few tweaks could further amplify the paper's impact within the community.